# A Data-Mining Interpretation Method of Pavement Dynamic Response Signal by Combining DBSCAN and Findpeaks Function

**DOI:** 10.3390/s24030939

**Published:** 2024-01-31

**Authors:** Hailong Liu, Ruqing Yao, Chunyi Cui, Jiuye Zhao

**Affiliations:** 1College of Transportation Engineering, Dalian Maritime University, Dalian 116026, China; liuhailong@dlmu.edu.cn (H.L.); zhaojiuye@dlmu.edu.cn (J.Z.); 2Center for Port and Maritime Safety, Dalian Maritime University, Dalian 116026, China; yaoruqing@dlmu.edu.cn; 3College of International Collaboration, Dalian Maritime University, Dalian 116026, China

**Keywords:** dynamic response of pavement, abnormal data diagnosis, adaptive feature extraction, DBSCAN algorithm, findpeaks function

## Abstract

During a heavy traffic flow featuring a substantial number of vehicles, the data reflecting the strain response of asphalt pavement under the vehicle load exhibit notable fluctuations with abnormal values, which can be attributed to the complex operating environment. Thus, there is a need to create a real-time anomalous-data diagnosis system which could effectively extract dynamic strain features, such as peak values and peak separation from the large amount of data. This paper presents a dynamic response signal data analysis method that utilizes the DBSCAN clustering algorithm and the findpeaks function. This method is designed to analyze data collected by sensors installed within the pavement. The first step involves denoising the data using low-pass filters and other techniques. Subsequently, the DBSCAN algorithm, which has been improved using the K-Dist method, is used to diagnose abnormal data after denoising. The refined findpeaks function is further implemented to carry out the adaptive feature extraction of the denoised data which is free from anomalies. The enhanced DBSCAN algorithm is tested via simulation and illustrates its effectiveness while detecting abnormal data in the road dynamic response signal. The findpeaks function enables the relatively accurate identification of peak values, thus leading to the identification of strain signal peaks of complex multi-axle lorries. This study is valuable for efficient data processing and effective information utilization in pavement monitoring.

## 1. Introduction

With the rapid growth of digitalization, road in the future will be transformed to be a kind of smart infrastructure. The rapid advancements in big data, artificial intelligence, and remote sensing technology have established the foundation for the intelligent monitoring of pavement behavior. Over the past few years, a substantial growth in private vehicle ownership has been observed. Due to the significant rise in the movement of heavy vehicles, numerous asphalt pavements have shown a trend of premature deterioration. This will have an adverse effect on driving comfortability and minimize the service life of asphalt pavements. In previous research [1,2,3,4], some researchers employed macroscopic approaches to assess pavement performance under traffic loads. For instance, a combination of high-definition cameras and digital-image processing methods can be employed to detect pavement damage and assess surface texture. Alternatively, a falling weight deflectometer and surface deflection screening can be used to measure road surface deflections and perform modulus back-calculations; however, these techniques are not only time-consuming and labor-intensive but also incapable of offering the long-term real-time monitoring of pavement infrastructure. Sensors embedded within the asphalt pavement can monitor the stress–strain responses of each structural layer in real time to provide the essential data for the precise evaluation of the service performance of asphalt pavement [5,6]. Nevertheless, the service environment can be very harsh for the sensors embedded inside the road’s surface. During long-term vehicle loading, the data collected by the embedded sensors in complex environments may display considerable fluctuations and data outliers. This will cause trouble for the subsequent work, such as the real-time extraction of feature peaks from large amounts of datasets.

Abnormal data generally refers to a deviation in the normal data of a dataset, which occurs due to its faults or the influence of the surrounding environment [7,8]. The traditional diagnosis of abnormal data generally requires subjective awareness as well as a considerable amount of time and effort in human identification. The manual screening of abnormal data is greatly subjective, which can lead to bias in assessing the service condition of roads [9,10]. Recently, an increasing number of scholars have conducted research on the automated diagnosis of abnormal data. For example, Luo et al. [11] used a Poisson-equation-based eigenvalue extraction model to extract data eigenvalues and then performed anomalous data detection using a K-Means model. This study achieved a correct detection rate of 98.72% with the implementation of the Poisson equation algorithm. Saeed et al. [12] proposed a 3-D deployment-based underwater anomaly localization network to detect anomalous anchor points. This network combines a 3-D localization technique with target distance localization and outlier anchor detection, which could effectively improve the localization accuracy of underwater sensors. Liu et al. [13] clustered electricity consumption and fluctuation data separately using the Mean-Shift clustering algorithm. After deriving the anomalous values from both datasets, they fed them into an XGBoost decision tree model for automatic learning and the classification of users with abnormal electricity consumption. Krishna Narayanan et al. [14] proposed a new deep learning approach for detecting anomalous behavior in sensor-based cyber-physical systems (CPSs) using a deep belief network (DBN) framework. This study also removed noisy data after the preprocessing. Kong et al. [15] proposed a framework for urban hierarchical anomaly detection. Based on the rough characteristics of traffic flow anomalies, they predicted traffic flow using a long short-term memory (LSTM) network and compared it with actual data to obtain historical anomaly scores. Anomaly area detection was subsequently performed using a one-class support vector machine (one-class SVM), and the traffic flow within the target and neighboring areas was analyzed from multiple perspectives to verify the effectiveness of the proposed method.

The methods mentioned above for diagnosing anomalous data in signals are widely distributed across various fields. The anomalous data diagnosis involves complex machine learning models and clustering-related algorithms. While the types of data targeted are various since they are collected by different road sensors [16,17,18], the main function of the aforementioned methods is to automatically classify data, which is currently the most common approach for abnormal data diagnosis. The DBSCAN clustering algorithm is a representative density-based clustering method that utilizes a set of neighborhood parameters (Eps and MinPts) to define the set’s sample density [19,20]. This will lead to the division of sample data into distinct clusters, the identification of noise as well as the detection of outliers.

Numerous studies have focused on signal peak identification in recent years. For instance, Zhang et al. [21] proposed a cascaded convolutional neural network (CNN) algorithm for fusion target detection to address the problem of peak point localization of quantitative fluorescence immunochromatography images being easily affected by multiple factors. The experiments showed that the average accuracy of the algorithm for peak point localization in these images could reach over 96%. Li et al. [22] proposed a joint wavelet identification peak point method based on the principle of wavelet multi-resolution analysis to correct baseline interference affecting the amplitude of the signal’s peak point. Then, they used the quadratic spline wavelet modulus maxima algorithm to automatically identify the peak point.

Research on signal peak extraction is primarily conducted using image-processing techniques, as well as signal data decomposition and reconstruction [23,24]. The accuracy of signal peaks may be difficult to guarantee due to significant differences in data format requirements and algorithm complexity [25]. Consequently, the applicability of peak extraction methods for road sensor signal data is limited. For the two-dimensional time series data collected by road sensors, the built-in Matlab function, findpeaks, may offer improved performance in extracting peaks. However, the algorithm requires improvement for peak detection in complex signal data scenarios [26].

To fully utilize the vast data collected by road sensors, this paper first diagnoses the data using the DBSCAN clustering algorithm for abnormal data detection and adopts the K-Dist descending ranking method to determine the important parameters within the algorithm. Using this method, the anomalies that cannot be removed by traditional noise reduction methods can be effectively identified, which largely improves the efficiency of that anomalous data diagnosis. This enables the unsupervised automatic detection of abnormal data in pavement dynamic response signals. Subsequently, the paper proposes a peak detection algorithm based on the improved Matlab findpeaks function, which is intended for analyzing data after the detection of anomalies of pavement dynamic response signals. Based on this method, different types of peak and valley points can be determined, which could effectively evaluate the deformation of the road’s surface when it is crushed by vehicles.

## 2. Materials and Methods

### 2.1. Engineering Background and Data Sources

The algorithm in this paper was validated using original data from the Yellow River Avenue project in the Jinan pilot area of Shandong Province. Sensors were embedded at various structural points within the road during the initial stages of road construction. This was to monitor the real-time deformation of the asphalt pavement under vehicle load. The sensors used are strain sensors, which undergo a recoverable change in their internal structure when subjected to stress. The structural change leads to a change in the electrical signal output from the sensor, which reflects the deformation of the sensor and thus characterizes the deformed condition of the pavement. The signal output of the sensor is collected in real time by a data collector when a vehicle passes over the pavement, thus obtaining a large number of signal values that reflect the deformation of the pavement. The asphalt pavement in this road section is a semirigid base pavement, which is commonly used in China. The road is constructed with a 4 cm SMA-13 surface course, a 6 cm AC-20C middle course, and an 8 cm AC-25C base course, as shown in Figure 1a. The base and sub-base consist of cement-stabilized macadam, and the subgrade material is cemented soil. A sensor is embedded on the surface of the base to monitor the deformation of the entire surface under the traffic load. To collect surface course strain information accurately during the pass of vehicles, sensors are embedded near a single-wheel track belt in a lane with a spacing of 60 cm. Figure 1b,c illustrate the method for embedding the sensors. The data acquisition instruments are connected to the ground wires and shield cables, resulting in a basic noise fluctuation of only 1–3 με on the surface of the base course sensors.

A 6.8 m long twin-axle platform truck manufactured by Dongfeng Motor Corporation was used in the test to obtain the strain signal from the sensor under a standard axle load. Before the test, the exact location of the buried sensors was verified and marked by the rear wheels of the truck, as shown in Figure 2a. The inflation pressure of each tire is maintained at 1.2 MPa throughout the test. The axle load of the rear axle (single-axle dual-wheel set) is controlled at 10 t by the loading and unloading of standard sand, as shown in Figure 2b, while the axle load of the front axle (single-axle single-wheel set) is 2.8 t. The air pressure and axle load mass of the above truck tires were in accordance with the Chinese norms JTG D50-2017 and the actual situation. During the calibration test, the vehicle passes a control point at different speeds (10, 20, 30, and 40 km/h) to apply a dynamic load. At the same time, a high-frequency data acquisition system collects the road dynamics signals from the sensors.

The dynamic load test was conducted between 14:00 and 17:00 (Beijing Time). The weather conditions at the site were cloudy. The direct solar radiation was not strong, and the atmospheric temperature was around 29 °C (84.2 °F). Throughout the test, the temperature recorded at the base of the pavement ranged from 29.3 °C (84.7 °F) to 29.6 °C (85.3 °F), while the temperature recorded at the top of the pavement ranged from 37.1 °C (98.8 °F) to 37.9 °C (100.2 °F). These results suggest that the temperature difference within the layers of the pavement structure was insignificant.

Figure 3a displays the original signal of the dynamic response of a road sensor under a dynamic vehicle load. The oscillogram shows significant fluctuations in the collected dynamic response signal, which was caused by environmental influences. The three peaks observed in the data correspond to the response signals when the vehicle’s three axles individually pass over the sensors. The presence of noise in the peak area poses challenges in accurately extracting and recognizing these peaks. Figure 3b presents the data after wavelet denoising, where the three peaks are visible; however, extracting the peak values manually is not an efficient process. Thus, this paper focuses on investigating automatic recognition and extraction methods for signal peaks. Furthermore, the paper also aims to facilitate the fine diagnosis of abnormal values in road dynamic response data.

### 2.2. Abnormal Data Diagnosis of Pavement Dynamic Response Signal Based on DBSCAN Algorithm

#### 2.2.1. Introduction to DBSCAN Algorithm

The DBSCAN algorithm is a typical density-based unsupervised clustering algorithm. It can divide the sample data into different clusters based on the affinity of the sample set, which could identify noise values and facilitate outlier detection. In the clustering process, the sample data is divided into three categories: core points, border points, and noise (as shown in Figure 4). A core point refers to a sample point whose number of data samples within the given Eps radius is greater than or equal to MinPts. A border point refers to a sample point where the number of data samples within the given Eps is less than MinPts but is located within the Eps radius of another core point. Noise refers to data samples that are not divided into any cluster [27].

The DBSCAN algorithm uses two input parameters (Eps and MinPts) to cluster the sample data. It is mainly based on the definitions of directly density-reachable and density-reachable. Directly density-reachable means a sample point *x_j_* is directly density-reachable from a core point *x_i_* if *x_j_* is within the Eps neighborhood of *x_i_*, while density-reachable denotes that if a series of sample points *p_1_*, *p_2_*, …, *p_n_* exists such that each point pi is directly density-reachable from the point *p_i__−_
_1_*, then *p_n_* is density-reachable from p1. This involves dividing all data points that are density-reachable from the same core point into the same cluster, aiming to find clusters of any shape and maximizing the similarity of samples within each cluster [28]. When the algorithm is used for anomalous data detection, the detected abnormal data is a by-product of the clustering process, namely the noise points. In this paper, by using road dynamic response monitoring data, the algorithm divides only the normal data into a cluster and identifies noise points as abnormal values.

#### 2.2.2. Determination of Eps and MinPts

Although the DBSCAN algorithm can cluster data into clusters of different shapes and screen out abnormal data, the Eps and MinPts parameters have a significant impact on the clustering effect. For Eps, if the value is too large, some less obvious noises points cannot be identified. While if the value is too small, many normal data points will be identified as noise, leading to deviation in the data analysis. For MinPts, this value is usually fixed during the algorithm’s execution and can be determined based on experience [29].

In conformity with the characteristics of the DBSCAN algorithm, the K-Dist descending curve method is adopted to determine Eps. First, based on the data characteristics of the road dynamic response, the selected MinPts values were 3, 4, 5, 6, and 7. Using the K-Dist method, the distance between each sample point and its K-nearest point is calculated. Additionally, these distances are then sorted from the largest to the smallest for plotting. The value of K is the same as that of MinPts. Figure 5 shows the descending order of the distance between each sample point and its third nearest point. From the graph, it can be concluded that the distance between most sample points and their third nearest point is about 2. When the distance value is between 2.0 and 2.15, specifically at the position marked by the arrow in the figure, the curvature is the steepest. In this interval, the distance value is too large, indicating that the abnormal data is far from the normal data. Therefore, it can be deduced that the optimal Eps parameter for the DBSCAN algorithm is between 2.0 and 2.15. In this study, the optimal Eps is set to 2.08 and can be fine-tuned to the appropriate units as required.

Through the analysis of the data collected at an earlier stage, it can be concluded that the density of data samples of the road dynamic response signals collected by different road sensors varies from each other. Therefore, when using the DBSCAN algorithm to diagnose abnormal data, it is necessary to use the aforementioned method to determine the Eps value for each sensor. The MinPts and K values can be slightly adjusted based on the data characteristics. If the data collected by the sensor remains stable over a long period, the Eps value does not need to be changed. Otherwise, the Eps value needs to be calibrated regularly.

#### 2.2.3. Implementation and Simulation of DBSCAN Improved Algorithm

Addressing the limitations of the traditional DBSCAN algorithm in abnormal data diagnosis, this study primarily optimizes the selection of the Eps parameter which aids in the diagnosis of abnormal values in the data collected by various types of sensors. The flow chart of abnormal road dynamic response data diagnosis process is shown in Figure 6.

The data collected by road monitoring sensors are a time series. General abnormalities, such as a single-strain value fluctuating slightly above or below the baseline, are typically cleared during the data denoising process; however, during the operation of the road sensor, the strain value at a certain time point may change dramatically, as shown in Figure 7. This is simulated in the abnormal data examples from the sensors above. These two types of abnormal data cannot be removed by the denoising process, indicating the significance of the use of the DBSCAN algorithm for further diagnosis.

In this paper, to verify the algorithm’s reliability, pavement dynamic response data from three different sensors were collected. Eight abnormal data points were randomly introduced post-noise reduction, and the Eps values were determined using the K-Dist descending plotting method. Subsequently, the DBSCAN algorithm implemented in Matlab R2023b was used for the abnormal data diagnosis.

As shown in Figure 8a, eight abnormal data points were randomly introduced into the data collected by Sensor I post-noise reduction. An analysis of the figure reveals two primary types of abnormalities: large-strain abrupt change points and small-strain abnormal values. Following the K-Dist descending sorting process, the Eps value ranged from 3.0 to 4.0. After fine-tuning, the Eps value was set at 3.25 (as shown in Figure 8b). The selected parameter was then incorporated into the DBSCAN model, and the corresponding diagnostic result is displayed in Figure 8c. Following the clustering analysis, seven out of the eight abnormal data points were successfully diagnosed. All large mutation points were also detected. The undetected anomalous points were attributed to their proximity to the main data baseline, indicating the need for further parameter adjustments.

For Sensor II, all eight abnormal data points, including those very close to the main curve, were successfully diagnosed, as shown in Figure 9. This indicates that the DBSCAN density clustering utilizing the K-Dist method for the Eps parameter calibration demonstrates high diagnostic accuracy for abnormal data from the pavement sensors.

As Figure 10 shows, a large difference in data density was observed in the data collected by Sensor III. The data density decreased significantly during the formation of the two peaks, which could have influenced the diagnosis of the abnormal data; however, after fine-tuning the Eps parameter, the detection results met the requirements.

According to the aforementioned analysis, the DBSCAN clustering algorithm can effectively detect all significant abnormal mutations. Moreover, the algorithm also demonstrates a good diagnostic effect on minor abnormal mutations that are typically removed by noise reduction. Therefore, the improved DBSCAN algorithm presented in this paper is reliable for diagnosing anomalies in pavement dynamic response data.

## 3. Results and Discussion

### 3.1. Automatic Peak Finding of Pavement Dynamic Response Signals Based on the Findpeaks Function

After removing the abnormal data, the strain peak eigenvalues are extracted using the findpeaks function. This peak information subsequently facilitates the analysis of pavement fatigue and deformation, as well as vehicle speed, axle number, and other factors.

The findpeaks function is a built-in feature coded in Matlab. Its basic principle involves comparing the strain value of a point with its neighboring strain values. If a point’s strain value is larger than those of its neighbors, it is identified as a peak. This function could swiftly identify the peak strain in the dynamic response signal of the pavement and its corresponding time point. However, relying solely on this basic function to identify peaks has proved to be insufficient for the data collected by the sensor in this project. This is due to the fluctuating nature of the data collected by the road sensors, where even minor changes might be misidentified as peaks. Furthermore, this function can identify peaks but not troughs. Since the pavement’s structural layer experiences both tension and compression under vehicle load, the sensor displays positive peak values when under tension and negative valley values when under compression (as shown in Figure 7); therefore, it is necessary to improve the findpeaks function before its application.

### 3.2. Findpeaks Function Peak Search Method Improvement

The findpeaks function is not only used for identifying peaks based on its principal mechanism but can also be configured with key parameters for detecting peaks in complex situations (such as MinPeakHeight, MinPeakDistance, MinPeakWidth, MaxPeakWidth, MinPeakProminence, and a specified minimum-adjacent difference).

To address the issue of small data fluctuations being identified as peaks, MinPeakHeight was initially chosen as the peak-seeking criterion to set a threshold value. Peak points below this threshold value would be disregarded. As the data baseline is variable, selecting a fixed threshold for this purpose was not practical. Consequently, this paper presents a method for determining the MinPeakHeight threshold using normal distribution analyses. A specific confidence level was chosen, and then data samples were analyzed for normal distribution using Matlab’s normfit function to obtain the upper and lower limits of the confidence interval, as shown in Equations (1) and (2). The upper limit of this interval, denoted as “a”, was established as the minimum peak height threshold.
(1)$$a=X+\frac{\sigma }{\sqrt{n}}Z_{\alpha /2}$$
(2)$$b=X-\frac{\sigma }{\sqrt{n}}Z_{\alpha /2}$$
where *a* is the confidence upper limit, *b* is the lower confidence limit, *X* is the data sample mean, *σ* is the standard deviation of the data sample, *n* is the sample size, and *Z*_α/2_ is a constant determined by the confidence level.

When a peak fluctuates during data generation, this significantly impacts peak identification. As shown in Figure 11, such fluctuations can result in two points matching the MinPeakHeight criterion. To address this issue, MinPeakProminence is adopted to manage fluctuations in the data at the peak position.

The peak bulge is defined as the vertical distance from the peak point to the first significant rise in value during the descent from both sides of the peak, as illustrated in Figure 12. Each peak has two peak bulge values. In terms of MinPeakProminence, this criterion is applied when the signal traverses both sides of a peak and reaches a point higher than the peak itself; this ascertains whether the peak bulge exceeds the minimum peak bulge value or not. The minimum peak bulge value is typically determined empirically. In this paper, it is set at one-twentieth of the maximum difference observed in the data.

Since the findpeaks function cannot identify valley values, this paper implements the negation of the data, followed by applying the peak-seeking step to the negated data. The MinPeakHeight is set as the negation of the lower confidence limit b, and the peak identified in the negated data corresponds to the valley value.

After the above-mentioned improvement of the findpeaks function, the extraction steps for the peak eigenvalues of the dynamic response of the pavement in this paper are shown in Figure 13.

### 3.3. Analysis of Data Peak-Seeking Results

To verify the reliability of the peak-seeking algorithm presented in this paper, the algorithm was first applied to the dynamic response signals of a vehicle at various speeds during the road loading test.

Figure 14 displays the peak finding results for various vehicle speeds. The algorithm accurately identifies the time point of the peak and the strain peak, providing a database for the subsequent analysis of the road structure’s performance. Additionally, analyzing the time point of the peak’s occurrence can assist in determining the driving speed of the vehicle.

The vehicle used for this test is a 6.8 m two-axle flatbed truck manufactured by Dongfeng, with a wheelbase of 5.1 m. As demonstrated in Figure 14a, the time difference between the two axles passing over the sensor is 1.822 s, indicating a driving speed of approximately 10.08 km/h, consistent with the actual situation. In Figure 14b, the calculated speed is 19.78 km/h. Figure 14c reveals a calculated speed of 29.28 km/h, and Figure 14d indicates a speed of 42.90 km/h. The analysis demonstrates that the method’s peak-seeking accuracy for the road dynamic response signal is promising, and that the deviation between the calculated and actual vehicle speeds is minimal. The peak position coincides with the actual moment when the vehicle crosses the sensor, confirming the accuracy of the peak’s positioning. Moreover, the magnitude of the peaks found corresponds with the actual conditions, indicating that the obtained strain values reflect the true deformation of the pavement surface.

In addition to the previously mentioned simple peak scenarios, this paper also explored automatic peak finding in situations with complex data fluctuations and the multi-peak scenarios of multi-axle heavy duty trucks. As illustrated in Figure 15, the data collected by this sensor is very complex, with baseline fluctuations becoming more pronounced in the absence of a vehicle load, particularly between 4–5 microstrains. Data fluctuations in certain zones are more pronounced during peak formation under traffic load, as indicated by the arrows in Figure 15. For addressing baseline fluctuations, the MinPeakHeight parameter adopted in this paper is very effective, whereas the MinPeakProminence method is primarily used for peak line fluctuations.

Moreover, this paper also analyzes data collected by the sensor when a random heavy-load truck passed over it, and the peak-seeking results are presented in Figure 16. This scenario is more complex than that in Figure 15 as it exhibits a greater number of peaks. The peak-seeking results reveal a total of six peaks, suggesting that the vehicle is a six-axle semi-trailer truck, which is consistent with field observations.

## 4. Conclusions

This paper analyzes the characteristics of dynamic response signals from road surfaces collected by embedded sensors. Firstly, an automatic detection method based on DBSCAN for road sensor data is established. The framework and workflow of this detection method are also introduced. Then, a simulation detection test using the field loading test data is conducted. Subsequently, Matlab’s findpeaks function is improved to automatically identify peaks in complex data fluctuation scenarios and multi-peak situations involving multi-axle heavy-duty trucks. The following conclusions could be drawn:(1)To address the sensitivity of the DBSCAN clustering algorithm to the Eps parameter, this parameter is determined through a K-Dist descending sorting process on the collected data. This could effectively diagnose large, abrupt abnormal values in the data.(2)The improved DBSCAN algorithm can detect most outliers, especially those close to the main data curve. This demonstrates the high diagnostic accuracy of DBSCAN density clustering, using the K-Dist method for Eps parameter calibration in identifying outlier data from pavement sensors.(3)The improved findpeaks function could accurately identify the time points and strain peaks, avoiding errors from data fluctuations in the baseline and peak formation process.(4)Data from the sensors could also capture a random heavy load truck and exhibit both baseline and peak formation fluctuations. Although these data are characterized by numerous peaks, the peak-finding algorithm can accurately identify the strain peaks.

Although this paper focused on the feasibility and accuracy of the algorithm, the running speed of the program is not fully optimized in the face of the huge amount of data. So a follow-up study is still needed, particularly for improving the running speed of the procedure, the integration of the procedure, and other aspects.

## Figures and Tables

**Figure 1 sensors-24-00939-f001:**
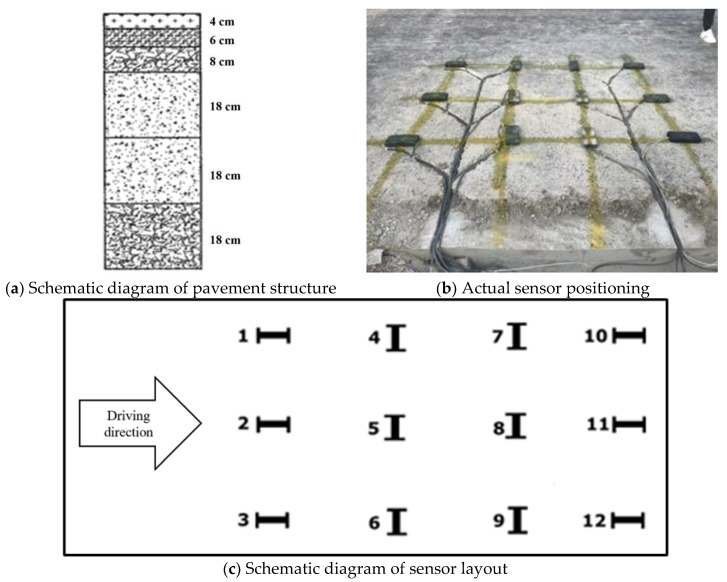
Schematic diagrams of pavement structure and sensor layout.

**Figure 2 sensors-24-00939-f002:**
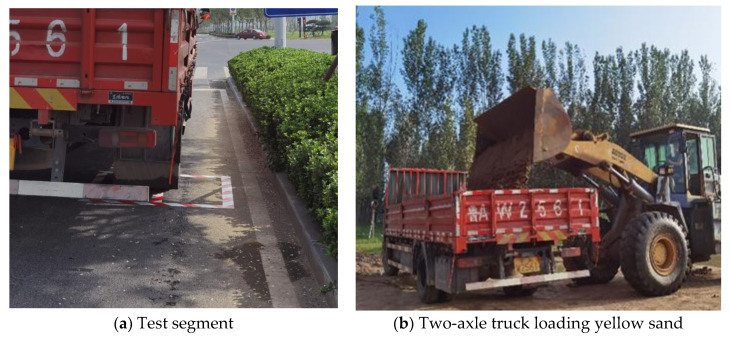
Test segment of the loading test and equipment.

**Figure 3 sensors-24-00939-f003:**
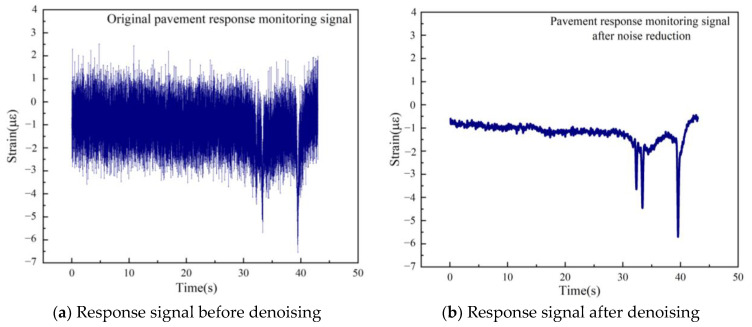
Signal time domain diagram.

**Figure 4 sensors-24-00939-f004:**
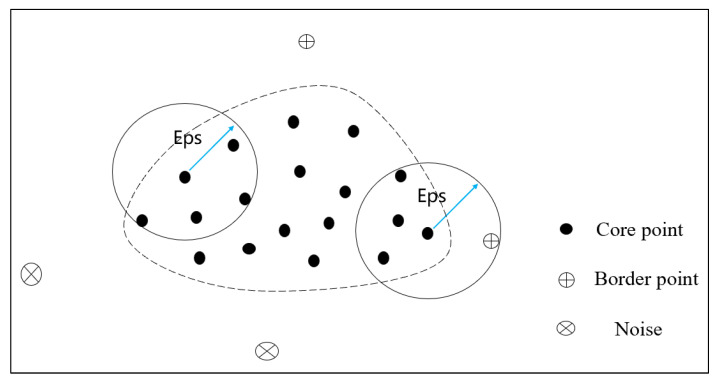
Schematic diagram of different data types in DBSCAN algorithm (MinPts = 3).

**Figure 5 sensors-24-00939-f005:**
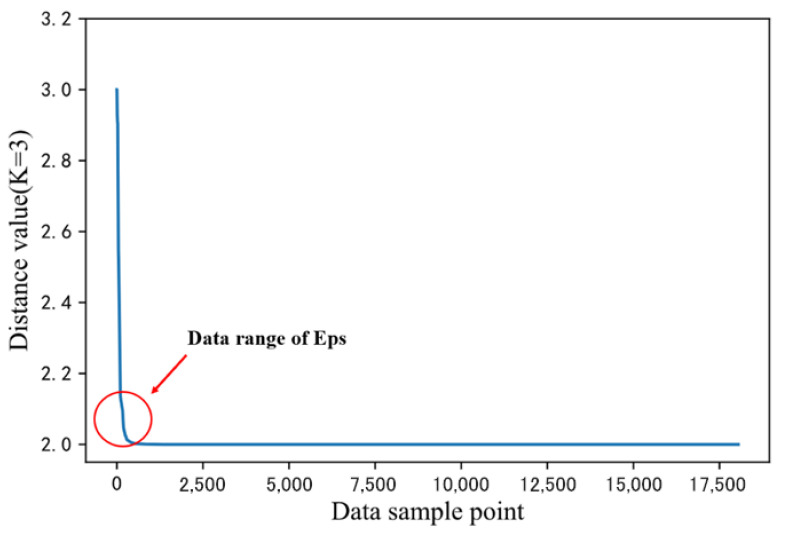
K-Dist descending curve.

**Figure 6 sensors-24-00939-f006:**
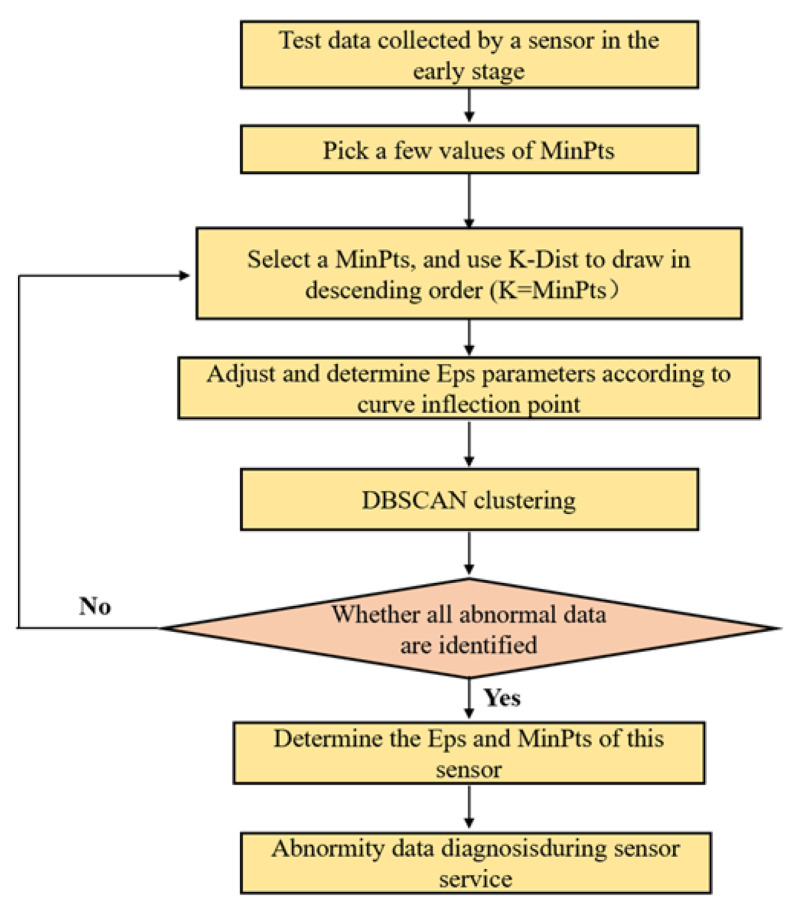
Flow chart of abnormal road dynamic response data diagnosis based on DBSCAN.

**Figure 7 sensors-24-00939-f007:**
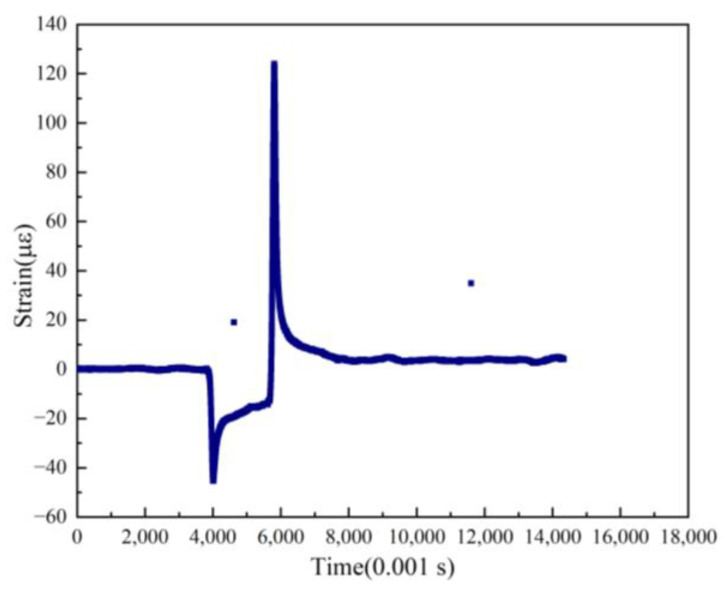
Schematic diagram of large mutation anomaly points.

**Figure 8 sensors-24-00939-f008:**
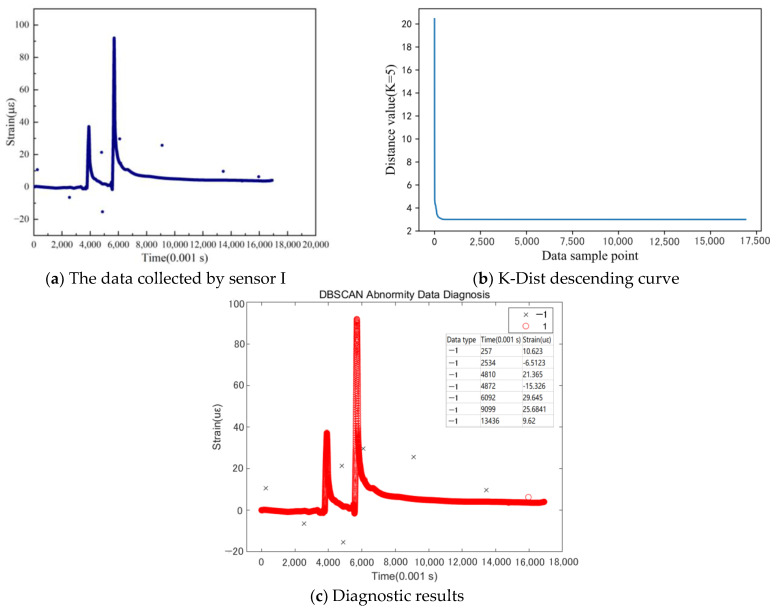
Abnormal data diagnosis of sensor I (K = MinPts = 5, Eps = 3.25).

**Figure 9 sensors-24-00939-f009:**
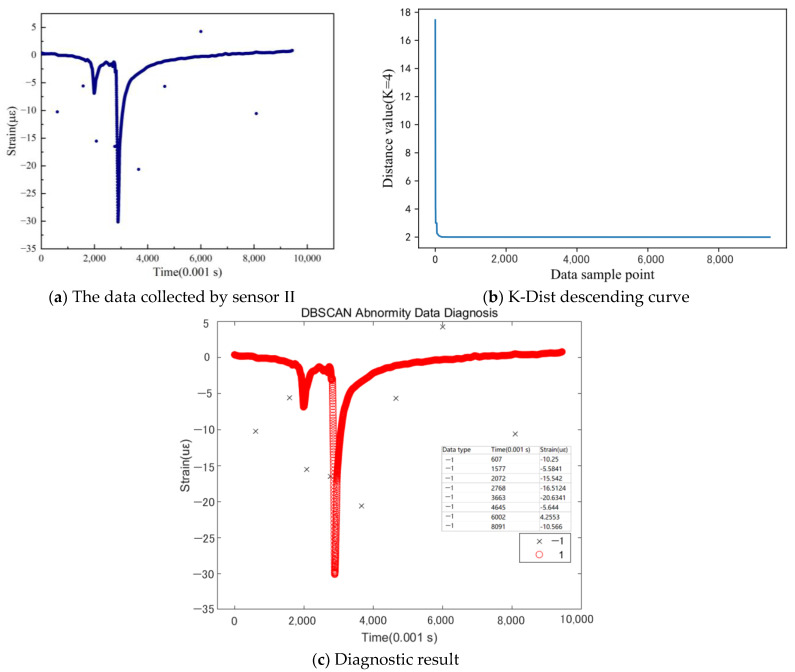
Abnormal data diagnosis of sensor II (K = MinPts = 4, Eps = 3.0).

**Figure 10 sensors-24-00939-f010:**
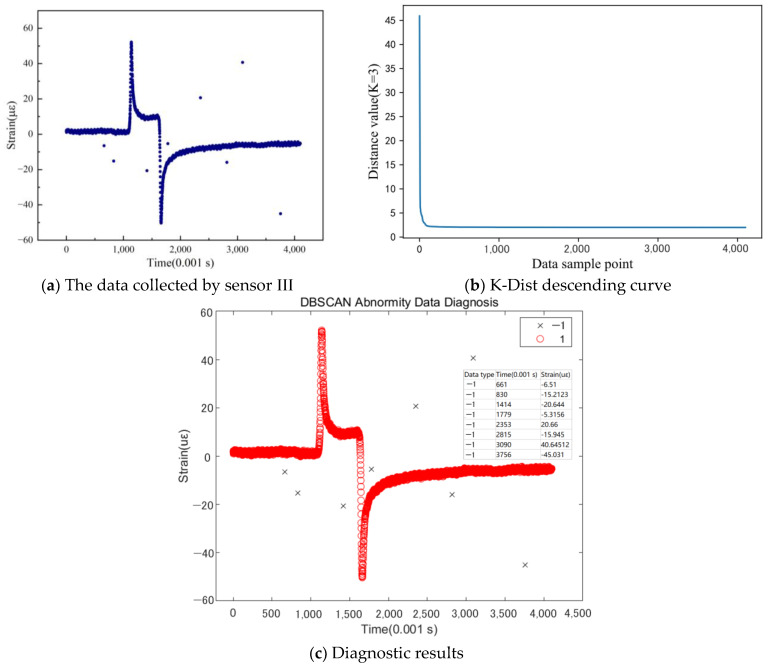
Abnormity data diagnosis of sensor III (K = MinPts = 3, Eps = 4.0).

**Figure 11 sensors-24-00939-f011:**
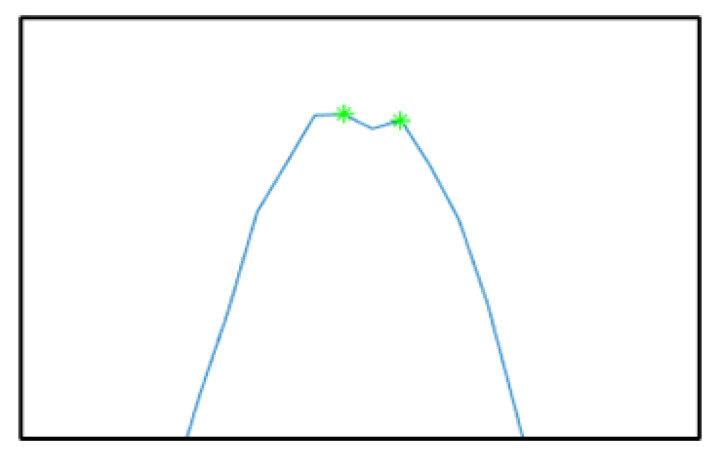
Schematic diagram of peak misclassification due to data fluctuation.

**Figure 12 sensors-24-00939-f012:**
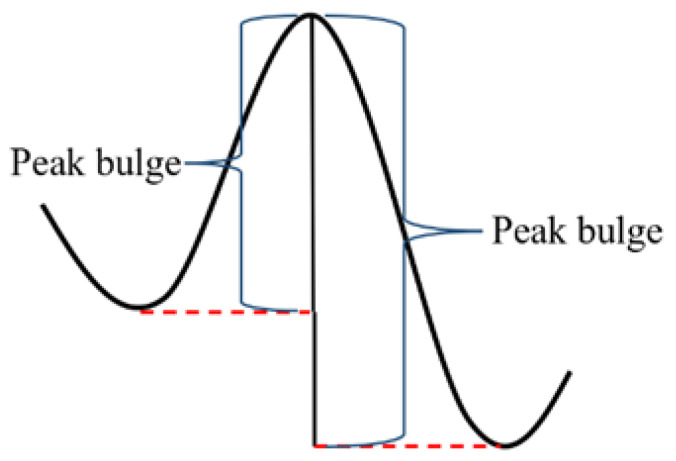
Schematic diagram of peak bulge.

**Figure 13 sensors-24-00939-f013:**
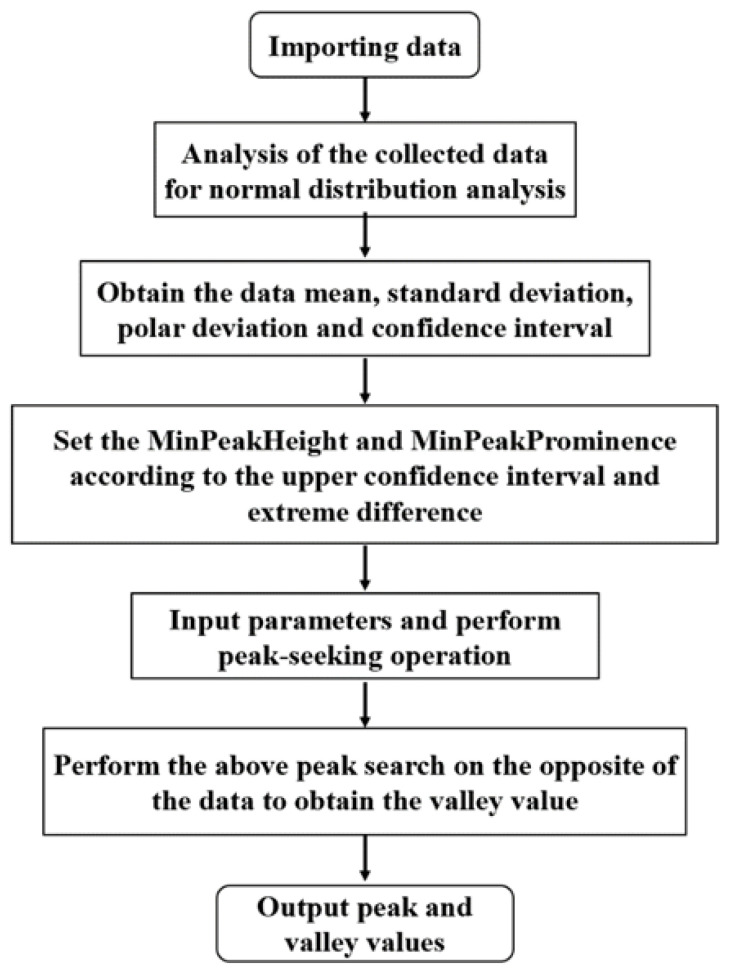
Schematic diagram of the peak eigenvalue extraction steps.

**Figure 14 sensors-24-00939-f014:**
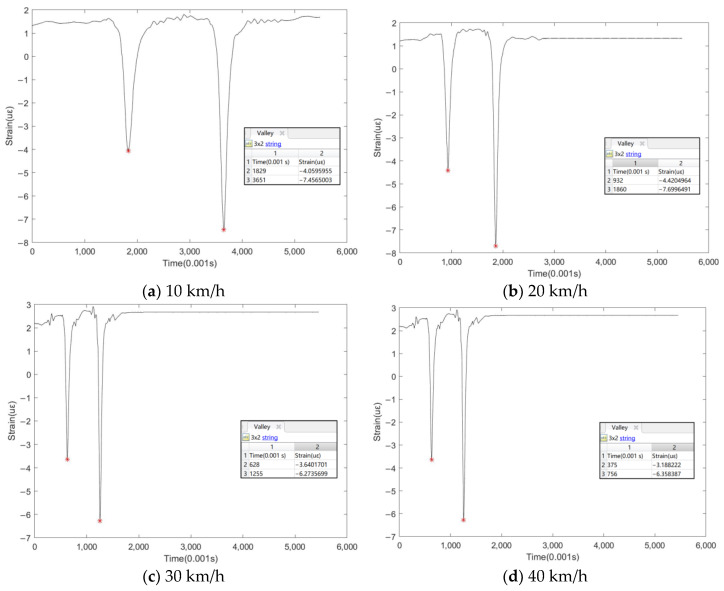
Peak-seeking signal of road dynamic response at different vehicle speeds.

**Figure 15 sensors-24-00939-f015:**
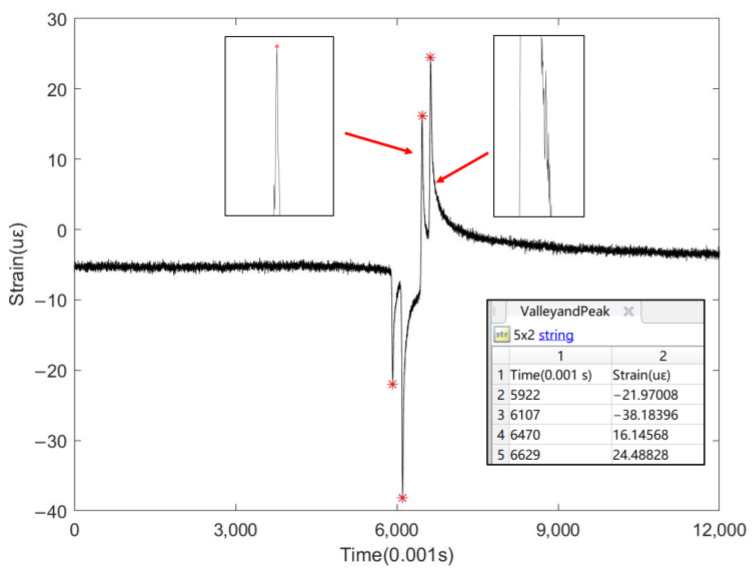
Automatic peak-seeking diagram under complex data fluctuations.

**Figure 16 sensors-24-00939-f016:**
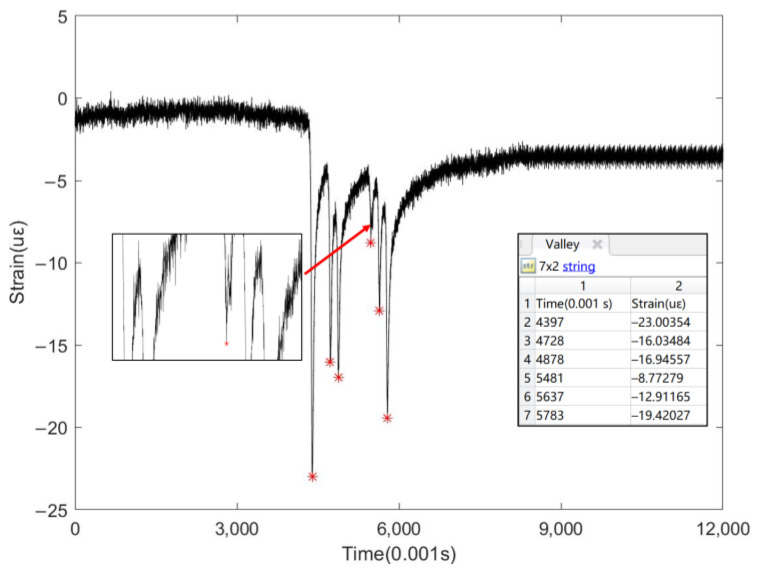
Automatic peak search function for multi-axle heavy-duty truck.

## Data Availability

The data reported in this article are available from the corresponding author upon request.

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
