# Peer review of "A Data-Mining Interpretation Method of Pavement Dynamic Response Signal by Combining DBSCAN and Findpeaks Function"

_sensors, 2024, doi:10.3390/s24030939_

Round 1

Reviewer 1 Report

Comments and Suggestions for Authors

1. In Introduction Line 80-81 need to supported by references.

2. In Introduction Line 100-105 need to supoort by references.

3. In Line 122 you did mention Fig1A.

4. Line 135 need to mention Fig2a and 2b, with more details.

5. You mention (The inflation pressure of each tire is maintained at 1.2 MPa throughout the test. The axle load of the rear axle (single axle dual wheel set) is controlled at 10t by loading and unloading yellow sand, while the axle load of the front axle (single axle single wheel set) is 2.8t. During the calibration test, the vehicle passes a control point at different speeds (10, 20, 30, 40 km/h) to apply a dynamic load. At the same time, a high-frequency data acquisition system collects the road dynamics signals provided by the sensors. ) Why select these testing condition? Need to add more details.

6. In Line 150-151 (These results suggest that the temperature difference within the layers of the pavement structure was minimal, likely due to the cloudy weather.) How the result can suggest ? Is there any stand for temperature level?

7. Fig.3 You need to discuss which method use to remove this noise.

8. I found few result in Section two and some part of Method in result and discussion.

9. Still need to clear show how you collect these data and noise reduction which can led for abnormal data. 

Comments on the Quality of English Language

English very difficult to understand/incomprehensible

Reviewer 2 Report

Comments and Suggestions for Authors

The authors present a dynamic response signal data analysis method that utilizes the DBSCAN clustering algorithm and the find peaks function to analyze data collected by sensors installed on the road. The methodology was described comprehensively and the results are meaningful and valuable. However, there are some problems existing in this paper that the authors must pay attention to deal with.

1.       The background and significance of this study should be highlighted in the introduction.

2.       Some words are too small in many figures to be recognized.

3.       Please provide the specific material preparation process.

4.       Some expressions in the manuscript are not clear. An English native speaker is suggested to carefully proofread it again.

5.       What is the meaning of baseline?

Comments on the Quality of English Language

 Some expressions in the manuscript are not clear. An English native speaker is suggested to carefully proofread it again.

Reviewer 3 Report

Comments and Suggestions for Authors

The paper deals with an interesting topic. The detection of the strain values provides valuable information for pavement assessment. My comments are the following:

page 1, line 42 instead of surface bending write surface deflections. It is a more accurate term. 

Please mention the limitations of the research and/or the research findings. Future investigation aspects should be also mentioned. 

page 1, line 32 "moni-toring" please delete the dash. There are a few other words in the text with the same problem. Please review the text carefully and correct them. 

In the conclusions section authors should comment on how their results can be used in terms of pavement monitoring or performance or maintenance. 

Comments on the Quality of English Language

Some words are separated with a dash. A correction regarding a term is suggested. The text should be read carefully to eliminate these issues. 

Round 2

Reviewer 1 Report

Comments and Suggestions for Authors

No more comments

Reviewer 2 Report

Comments and Suggestions for Authors

No comments